# Improved Hygroscopicity and Bioavailability of Solid Dispersion of Red Ginseng Extract with Silicon Dioxide

**DOI:** 10.3390/pharmaceutics13071022

**Published:** 2021-07-04

**Authors:** Sojeong Jin, Chul Haeng Lee, Dong Yu Lim, Jaehyeok Lee, Soo-Jin Park, Im-Sook Song, Min-Koo Choi

**Affiliations:** 1College of Pharmacy, Dankook University, Cheon-an 31116, Korea; astraea327@naver.com (S.J.); hang1130@naver.com (C.H.L.); twins3639@naver.com (D.Y.L.); 2BK21 FOUR Community-Based Intelligent Novel Drug Discovery Education Unit, Vessel-Organ Interaction Research Center (VOICE), Research Institute of Pharmaceutical Sciences, College of Pharmacy, Kyungpook National University, Daegu 41566, Korea; here0723@gmail.com; 3College of Korean Medicine, Daegu Haany University, Daegu 38610, Korea

**Keywords:** Korean red ginseng extract, solid dispersion formulation, hygroscopicity, intestinal permeability, oral bioavailability

## Abstract

This study aims to develop a powder formulation for the Korean red ginseng extract (RGE) and to evaluate its in vitro and in vivo formulation characteristics. The solid dispersion of RGE was prepared with hydrophilic carriers using a freeze-drying method. After conducting the water sorption–desorption isothermogram (relative humidity between 30 and 70% RH), differential scanning calorimetry thermal behavior, dissolution test, and intestinal permeation study, a solid dispersion formulation of RGE and silicon dioxide (RGE-SiO_2_) was selected. RGE-SiO_2_ formulation increased intestinal permeability of ginsenoside Rb1 (GRb1), GRb2, GRc, and GRd by 1.6-fold in rat jejunal segments as measured by the Ussing chamber system. A 1.6- to 1.8-fold increase in plasma exposure of GRb1, GRb2, GRc, and GRd in rats was observed following oral administration of RGE-SiO_2_ (375 mg/kg as RGE). No significant difference was observed in the time to reach maximum concentration (T_max_) and half-life in comparison to those in RGE administered rats (375 mg/kg). In conclusion, formulating solid dispersion of RGE with amorphous SiO_2_, the powder formulation of RGE was successfully formulated with improved hygroscopicity, increased intestinal permeability, and enhanced oral bioavailability and is therefore suitable for processing solid formulations of RGE product.

## 1. Introduction

A commonly used herbal supplement, red ginseng extract (RGE), has been investigated for its efficacy in many Asian countries, including Korea. It has been reported that RGE has several health benefits including reinforcing the immune system, having anti-cancer, anti-diabetes, anti-inflammation, and anti-oxidation properties as well as liver detoxification effects [1]. Ginsenoside, called ginseng saponin, has a sugar moiety structure bound to an oleanone and dammarane structure. Depending on the location of the carbon bond of the sugar chain, the type of dammarane ginsenoside is determined, either 20(*S*)-protopanaxadiol (PPD)-type or 20(*S*)-protopanaxatriol (PPT)-type (Figure 1). Further subdivisions occur depending on the length of the sugar chain. Belonging to the PPD-type ginsenosides are 20(*S*)-ginsenoside Rb1 (GRb1), GRb2, GRc, GRd, GRh2, GF2, GRh2, Compound K (CK), and PPD; whereas GRe, GRg1, GRf, GF1, GRh1, and PPT all belong to PPT-type ginsenosides (Figure 1). Human administration of large molecular weight ginsenosides (they have long sugar chains such as GRb1, Grb2, GRc, GRd, GRe, GRg1, and GRf) undergo deglycosylation by the intestinal microbiota to small molecular weight ginsenoside (have short sugar chains such as GRh2, CK, PPD, GRh1, GF1, and PPT) [2,3,4]. It is expected that the deglycosylated ginsenosides are likely to be more effective and permeable in the intestinal lumen [1,5].

It has been reported that ginseng and its associated ginsenosides have a range of properties including anti-neoplastic, anti-hypertensive, antidiabetic, anti-inflammatory, antioxidative, antiallergic, neuroprotective, hepatoprotective, and immunologic effects [6,7,8,9]. In addition to these therapeutic effects, ginseng products are frequently administered as health supplements with therapeutic drugs for improving fatigue and physical performance [10]. Therefore, the demand for ginseng products has grown rapidly, and a variety of ginseng products include fresh ginseng, white ginseng product, RGE, and others.

In Korea, the most frequently consumed product is RGE, and its diluted liquid formulation, according to the 2018 annual report by Korea Rural Economic Institute. The consumption of RGE in its solid formulation or tablet form has been limited due to the hygroscopicity issues of the dried RGE. Improving the hygroscopicity of RGE using a small volume of pharmaceutical excipients is a challenge for solid formulations of high-content RGE. It has been proposed that a microcapsule or coated granule of RGE with a waterproof polymer can be utilized (Patent No. 2003-0066136 and 2008-0015655). In addition, a tablet that improves the hygroscopicity of RGE by containing a mixture of sucrose fatty acid esters and monoglyceride has been proposed (Patent No. 2007-0118020 and 2010-0090038).

A common method used for powder formulation is solid dispersion, which involves mixing one or more main active ingredients with a carrier and subsequently drying it using melting, solvent evaporation, and solvent-melting methods to produce a solid powder. Advantages of the solid dispersion technique facilitate the use of this formulation. This powder becomes an amorphous solid powder and is an easy to formulate capsule or tablet using a variety of combinations of carrier excipients [11,12]. In addition, the solid dispersion technique is widely used for increasing oral bioavailability through the enhancement of its dissolution and intestinal permeability of the drugs [13]. The solid dispersion product can improve wettability and increase the surface area [14,15]. The solid dispersion technique also can resolve the hygroscopicity and reduce strong acid dissociation caused by salt formulation [15]. However, several disadvantages of solid dispersion should be pointed out. Regarding the physical instability of solid dispersion products, they are prone to be recrystallized by increasing the overall molecular mobility, decreasing glass transition temperature, or disrupting interaction between the drug and carrier, resulting in a decreased solubility and dissolution rate during the storage period [15,16]. Over the years, the number of polymers used as carriers in the solid dispersion products reported in the literature has grown remarkably [16]. Solid dispersion technology is not limited to the laboratory scale; it has expanded to commercial production [15,16]. In addition to the numerous investigational solid dispersions from the melting method (e.g., clotrimazole, fenofibrate, furosemide, paclitaxel, etc.), solvent evaporation method (e.g., dutasteride, tadalafil, glimepiride, nimodipine, etc.), hot-melt extrusion method (e.g., ritonavir, naproxen, efavirenz, tamoxifen, etc.), and freeze-drying method (e.g., nifedipine, celecoxib, docetaxel, etc.), a number of commercial solid dispersion products including Cesamet^®^ (nabilone-PVP; Lilly), Ceritan^®^ (everolimus-HPMC; Norvatis), Gris-PEG^®^ (griseofulvin-PEG; Norvatis), Intelence^®^ (etravirine-HPMC; Janssen), Adalat-XL^®^ (nifedipine-PEG3350/HPC/cellulose acetate; Bayer), Sporanox^®^ (itraconazole-HPMC; Janssen), Isoptin SR^®^ (verapamil-HPC/HPMC; Abbott), and Crestor^®^ (rosuvastatin-HPMC; AstraZeneca) have been marketed as tablet or capsule formulations [15,16].

Therefore, this technique could be applied to improve the hygroscopicity and to enhance the oral bioavailability of RGE. The oral bioavailability is generally low in a variety of ginsenosides (less than 5%) [5,10,17,18,19]. The bioavailability-improving strategies for ginsenosides have been applied as nanoparticles, liposomes, emulsions, and micelles [19,20]. The bioavailability of GR1, GRg1, and GRb1 in *Panax ginseng* saponin-loaded long circulating nanoparticles increased 3 to 4 times compared with those in *Panax ginseng* saponin [21]. The plasma exposure of GRg3 in liposomes prepared by polycarbonate membrane extrusion increased by 1.5-fold compared to GRg3 solution [22]. The absorption rates of GRh2 in self-microemulsion prepared by oleic acid ester, polysorbate 280, and transcutol P significantly increased compared with GRh2 itself [19]. Mixed micelle formulation with ascorbyl palmitate and α-tocopherol polyethylene glycol 1000 succinate monoester (10 ~ 30 nm) containing CK had the ability to increase cell uptake and tumor targeting [23]. Except for these lipid-based formulations, the solid dispersion formulation has never been reported. Therefore, in this study, we aimed to formulate a solid dispersion formulation of red ginseng extract that has high-adsorptive carriers to improve both the hygroscopicity and oral bioavailability of RGE.

## 2. Materials and Methods

### 2.1. Materials

RGE was purchased from Punggi Ginseng Cooperative Association (Youngjoo, Kyungpook, Korea) and was produced in the facilities following the current guidelines of the Korea Good Manufacturing Practice. This product (Hwangpoonjung, Lot No. 731902) contains more than 60% of dried ginseng cultivated for 6 years, and the total amount of GRb1, GRg1, and GRg3 marker ginsenosides is 9 mg/g RGE. The following ginsenosides were purchased from the Ambo Institute (Daejeon, Korea): 20(*S*)-ginsenosides Rb1 (GRb1), GRb2, GRc, GRd, GRg1, GRg3, GRe, GRh1, GF1, GF2, CK, PPD, and PPT (Figure 1). For the internal standard (IS), both Berberine and ^13^C-caffeine were used and purchased from Sigma-Aldrich Chemical Co. (St. Louis, MO, USA). Mannitol, lactose, sucrose, hydroxypropyl cellulose (HPC), and polyvinylpyrrolidone (PVP) were purchased from Sigma-Aldrich Chemical Co. (St. Louis, MO, USA). Trehalose, cross-linked polyvinylpyrrolidone (PVPc), polyethyleneglycol 1500 (P1500), and polyethyleneglycol 6000 (P6000) were provided by Acros (Leicestershire, England). Silicon dioxide (SiO_2_; Aerosil 200) was obtained from Dong-A Pharmaceutical Co., Ltd. (Seoul, Korea). All other chemicals and solvents utilized in this study were of reagent or analytical grade.

### 2.2. Preparation of Solid Dispersion of Red Ginseng Extract

To begin, 1 g of RGE was dissolved in 10 mL of water, and to this, 500 mg of a water-soluble carrier was added. Mannitol, sucrose, lactose, and trehalose were used as sugar moiety carriers; HPC, PVP, PVPc, P1500, P6000, and silicon dioxide were used as hydrophilic polymer carriers. The mixture was vigorously vortexed for 30 min using a Multi-reax stirrer (Heidolph, Schwabach, Germany), frozen for 12 h at −80 °C, and subsequently freeze-dried for 72 h using a freeze dryer (FDCF-12012, Operon, Gyeonggi-do, Korea). Samples were then ground using a mortar, passed through a Korean Pharmacopoeia sieve (mesh size = 150 μm), and stored in a desiccator until required for formulation characterization.

The ginsenoside content was measured in the solid dispersion formulations of RGE, RGE-SiO_2_ (RGE with silicon dioxide), RGE-M (RGE with mannitol), RGE-S (RGE with sucrose), RGE-L (RGE with lactose), RGE-T (RGE with trehalose), RGE-HPC, RGE-PVP, RGE-PVPc, RGE-P1500, and RGE-P6000. Briefly, the solid dispersion formulations of RGE alone (50 mg) and RGE with hydrophilic excipients (which were all equivalent to 50 mg RGE) were dissolved in 10 mL of distilled water on a rotary shaker at 60 rpm for 4 h. A 2 mL aliquot of a medium was filtered using a 0.45 μm membrane filter, and the concentrations of GRb1, GRb2, GRc, GRd, GRg1, GRg3, GRh2, GRe, GRh1, GF1, GF2, CK, PPD, and PPT in the filtrates were analyzed using a liquid chromatography-tandem mass spectrometry (LC-MS/MS) system.

### 2.3. Hygroscopicity Test

The hygroscopicity of the solid dispersion formulation of RGE was evaluated through visual observation, and the water content change was analyzed using a thermo-hygrostat incubator (JSCH-070CPL, JS Research Inc., Chungnam, Korea) with increasing humidity (from 30% RH to 70% RH) and decreasing the humidity (from 70% RH to 30% RH) at 30 °C for 50 days. Approximately 1g of solid dispersion formulations of RGE was placed on a sample tray under the corresponding relative humidity at 30 °C in a thermo-hygrostat incubator, and the subsequent weight was measured once equilibrium was reached (i.e., weight change < 0.1%).

Water sorption rate was calculated as follows [24,25]:(1)$$W\;\left(\%\right)=\frac{W2-W1}{W1}\times 100.$$
where *W* represents water sorption rate, and *W*1 and *W*2 represent the initial weight and the weight of the sample after exposure to humid conditions, respectively.

### 2.4. Characterization of RGE Solid Dispersion

Dissolution studies were conducted in 900 mL of distilled water for 240 min in a D-63150 dissolution test apparatus (Erweka, Heusenstamm, Germany) at 37 °C and 50 rpm using a paddle method (a type 2 USP dissolution method). Briefly, solid dispersion formulations of RGE alone (50 mg) and RGE with hydrophilic carriers (all are equivalent to 50 mg RGE) were packaged into a hard gelatin capsule (size No. 0) and placed inside a sinker. A 1 mL aliquot of a medium was collected at 0, 5, 10, 15, 20, 30, 60, 120, 180, and 240 min and filtered using a 0.45 μm membrane filter, with an equal volume of water replaced after each sampling. The concentrations of GRb1, GRb2, GRc, and GRd in the filtrates were analyzed using an LC-MS/MS system.

X-ray diffraction (XRD) of SiO_2_, mannitol, PEG6000, solid dispersion formulation of RGE, RGE-SiO_2_, RGE-M, and RGE-P6000 was determined on an Empyrean X-ray diffractometer (Malvern Panalytical Ltd., Malvern, England) using Cu Kα radiation, at 40 mA and 40 kV. Data were obtained from 5° to 70° (2θ) with a step size of 0.02°, and a scanning speed of 5°/min.

Differential scanning calorimetry (DSC) of SiO_2_, mannitol, PEG6000, solid dispersion formulation of RGE, RGE-SiO_2_, RGE-M, and RGE-P6000 was determined using a DSC Q2000 (TA Instruments, New Castle, DE, USA). For each sample, approximately 5 mg was placed into a closed aluminum pan and heated at a scanning rate of 5 °C/min from 10 °C to 250 °C, with nitrogen purging at 20 mL/min. The temperature scale was calibrated using indium.

### 2.5. Intestinal Permeability Test

Male Sprague Dawley rats (7–8 weeks old, weighing 225–270 g) were purchased from Samtako Co. (Osan, Korea). Rats were housed in a 12 h light/dark cycle, and food and water were supplied ad libitum for one week prior to animal studies. Rats were fasted for 16 h, but they had free access to water prior to the commencement of the study. Rats were subsequently anesthetized using isoflurane (isoflurane vaporizer to 2% with oxygen flow at 0.8 L/min). A proximal jejunum section (approximately 10 cm) was excised and washed in prewarmed HBSS (pH 7.4). Segments were mounted in a tissue holder of a Navicyte Easy Mount Ussing Chamber (Warner Instruments, Holliston, MA, USA), which had a surface area of 0.76 cm² and was acclimated in HBSS for 15 min with continuous oxygenation (95% O_2_ and 5% CO_2_ gas). Intestinal permeability studies began with the replacing of HBSS on both sides of intestinal segments with 1 mL of prewarmed HBSS containing RGE, RGE-SiO_2_, RGE-M, and RGE-P6000 (equivalent to 5 mg RGE) on the donor side, and 1 mL of prewarmed fresh HBSS on the receiver side. From the reliever side, 400 μL aliquots were removed every 30 min for 2 h, and an equal volume of prewarmed fresh HBSS was replaced on the receiver side. The concentrations of GRb1, GRb2, GRc, and GRd on the donor and receiver sides were analyzed using an LC-MS/MS system.

### 2.6. Pharmacokinetic Study

Male Sprague Dawley rats (7–8 weeks old, weighing 225–270 g) were fasted for 16 h with free access to water before the pharmacokinetic study. Rats were administered RGE (375 mg RGE/kg/2 mL in distilled water) and RGE-SiO_2_ (equivalent to 375 mg RGE/kg/2 mL in distilled water) via oral gavage. Blood samples were obtained from the cannulated femoral artery at 0, 0.5, 1, 2, 4, 8, 24, 30, and 48 h after the administration of the solid dispersion formulations of RGE. Blood samples were centrifuged at 16,000× *g* for 1 min, and 50 μL of plasma was stored at −80 °C until required for ginsenoside analysis.

### 2.7. Stability of RGE-SiO_2_ Solid Dispersion

XRD patterns of RGE-SiO_2_ stored in 30% RH at 25 °C for 1 year and freshly prepared RGE-SiO_2_ were also determined using an Empyrean X-ray diffractometer (Malvern Panalytical Ltd., Malvern, England).

DSC thermograms of RGE-SiO_2_ stored in 30% RH at 25 °C for 1 year and freshly prepared RGE-SiO_2_ were also compared using a DSC Q2000 (TA Instruments, New Castle, DE, USA).

The ginsenoside contents of RGE-SiO_2_ stored in 30% RH at 25 °C for 1 year and freshly prepared RGE-SiO_2_ were also measured by the same method described in Section 2.2 using the LC-MS/MS system.

Comparative dissolution studies with RGE-SiO_2_ (12 capsules filled in hard gelatin capsule (size No. 0); equivalent to 50 mg RGE), which was stored in 30% RH at 25 °C for 1 year, and freshly prepared RGE-SiO_2_ (12 capsules filled in hard gelatin capsule (size No. 0); equivalent to 50 mg RGE) were conducted in 900 mL of distilled water for 240 min in a D-63150 dissolution test apparatus (Erweka, Heusenstamm, Germany) at 37 °C and 50 rpm using a paddle method (a type 2 USP dissolution method). Experimental procedures were identical to the method described in Section 2.4. Using the mean dissolution values from both dissolution curves at 5, 10, 15, 20, 30, 60, 120, and 180 min, the similarity factor (*f*_2_) was calculated by the following equation [26]:(2)$$f_{2}=50\times \log\left(\frac{100}{\sqrt{\left\{1+\frac{1}{n}\sum_{t=1}^{n}{\left(R_{t}-T_{t}\right)}^{2}\right\}}}\right).$$
where *n* represents the number of time points, *R_t_* is the dissolution value of freshly prepared RGE-SiO_2_ solid dispersion at time *t*, and *T_t_* is the dissolution value of RGE-SiO_2_ solid dispersion stored in 30% RH at 25 °C for 1 year at time *t*. Time *t* was used from 5 to 180 min because the dissolution rates of ginsenosides GRB1, GRb2, and GRc were over 85% at 240 min in both groups. The dissolution profiles of ginsenoside from two different RGE-SiO_2_ formulations were considered to be similar when the *f*_2_ value was greater than 50 [26].

### 2.8. LC-MS/MS Analysis of Ginsenosides

The LC-MS/MS method utilized an Agilent 6470 triple quadrupole LC-MS/MS system (Agilent, Wilmington, DE, USA) to analyze the concentrations of ginsenosides [4,5,18]. Briefly, 400 μL of an IS (0.1 ng/mL berberine in methanol) was added to 50 μL of plasma samples, vortexed for 15 min, and subsequently centrifuged at 16,000× *g* for 5 min. After centrifugation, 200 μL of the supernatant was transferred to a clean tube, and a 4 μL aliquot was injected into the LC-MS/MS system. The ginsenosides were separated on a Polar RP column (150 × 2.0 mm, 4.0 μm particle size) (Phenomenex, Torrance, CA, USA) with a mobile phase comprising 0.1% formic acid in water (phase A) and 0.1% formic acid in methanol (phase B) at a flow rate of 0.3 mL/min. The gradient elution used was 70% of phase B for 0–0.2 min, 70–90% (phase B) for 0.2–1.0 min, 90% (phase B) for 1.0–6.5 min, 90–70% (phase B) for 6.5–7.0 min, and 70% (phase B) for 7.0–10.0 min. The column and autosampler temperatures were 40 °C and 6 °C, respectively. The electrospray ionization (ESI) source settings were as follows: gas temperature 300 °C; gas flow 10 L/min; nebulizer pressure 35 psi; capillary voltage 4000 V; and nozzle voltage 500 V. Quantification was performed using multiple reaction monitoring in the positive ion mode, and the details are shown in Table 1. For the 14 ginsenosides, the standard calibration curve for the mixture was linear in the concentration range of 0.5–200 ng/mL, and the inter-day and intra-day precision and accuracy for ginsenosides were <15%. The matrix effect and extraction recovery of the 14 ginsenosides using the methanol precipitation method had coefficients of variance <15%. No significant degradation was observed in the 14 ginsenosides from the short-term stability (4 h, 25 °C), post-treatment stability (6 °C, 24 h), and freeze–thaw cycle stability (−80 °C / 25 °C, 3 Cycles) measurements.

### 2.9. Data Analysis

Pharmacokinetic parameters were calculated using WinNonlin (version 5.1; Pharsights, Cary, NC, USA) with the non-compartmental analysis. The data are expressed as the means ± standard deviation for each group. Statistical analysis was performed using the Student *t*-test.

## 3. Results

### 3.1. Preparation of Solid Dispersion of RGE with Hydrophilic Carriers

To begin, using frequently used hydrophilic carriers, solid dispersion formulations of RGE were prepared and subsequently tested for the hygroscopicity of the RGE powder. The hydrophilic carriers were selected on the basis of their high dispersibility, increased solubility, and low hygroscopicity [11]. Saccharides (i.e., mannitol, sucrose, lactose, and trehalose), cellulose derivatives (i.e., HPC), polymers (i.e., PVP, and cross-linked PVP (PVPc)), and polyethylene glycols (i.e., P1500 and P6000) [11,27,28,29,30] were used in the preparation of solid dispersion formulations of RGE with low hygroscopicity and increased oral permeability. We began by determining the ginsenoside content in the solid dispersion formulations of RGE and whether the solid dispersion formulation process modified the ginsenoside content.

As shown in Table 2, no statistical difference was noted in the content of major ginsenosides in all solid dispersion formulations of RGE with hydrophilic carriers (i.e., RGE-SiO_2_, RGE-M, RGE-S, RGE-L, RGE-T, RGE-HPC, RGE-PVP, RGE-PVPc, RGE-P1500, and RGE-P6000) compared to RGE alone (*p* > 0.05). These results suggest that the solid dispersion manufacturing process with the 10 hydrophilic carriers used in this study did not affect the stability or content of the major PPD-type and PPT-type ginsenosides.

### 3.2. Hygroscopicity Test

The water sorption rate of the solid dispersion formulations of RGE was measured based on the effect relative humidity had after storing in a thermo-hygrostat at 30 °C with increasing the humidity from 30% RH to 70% RH (Figure 2). An increase in the water sorption rate of the solid dispersion formulations of RGE was observed with an increase in relative humidity; however, the increase in the water sorption rate was different depending on the carrier type. For example, the water sorption rate in the solid dispersion formulations of RGE with SiO_2_, mannitol, lactose, and P6000 was lower than 12%, whereas the water sorption rate of the solid dispersion formulations was higher than 17% for RGE, RGE-S, RGE-T, RGE-PVP, RGE-PVPc, and RGE-P1500.

In addition, the visual observations of the solid dispersion formulation of RGE with various hydrophilic carriers are shown in Figure 3. As with the data included in Figure 2, the solid dispersion formulations of RGE with SiO_2_, mannitol, PVPc, and P6000 had no observable differences after storage in various humid conditions. However, the use of sucrose, lactose, trehalose, PVP, and P1500 did not improve the hygroscopic nature of RGE powder. For formulations containing HPC and RGE-HPC, the solid dispersion did not formulate the powdery forms. The water sorption–desorption isotherms of various solid dispersion formulations of RGE at 30 °C are illustrated in Figure 4. The water sorption–desorption curves of solid dispersion formulations of RGE samples had different patterns according to the carrier types. The water sorption–desorption curves of RGE, RGE-S, RGE-L, RGE-T, RGE-PVP, and RGE-PVPc showed hysteresis; however, RGE-SiO_2_, RGE-M, RGE-HPC, RGE-P1500, and RGE-P6000 had no, or negligible, hysteresis. The hysteresis phenomenon between sorption phase and desorption phase during the relative humidity alteration indicated the irreversible change in water content and in the hygroscopic nature [31,32]. Utilizing the observations and data generated from the hygroscopicity test and water sorption–desorption curves of the solid dispersion formulations of RGE, three formulations were selected for further studies: RGE-SiO_2_, RGE-M, and RGE-P6000.

### 3.3. Characterization of RGE Solid Dispersion Formulation

#### 3.3.1. XRD and DSC Analysis 

To characterize the solid dispersion formulations of RGE-SiO_2_, RGE-M, and RGE-P6000, XRD patterns, the DSC thermal behavior, dissolution rate, and intestinal permeability were investigated and compared to RGE alone. XRD patterns of the hydrophilic carriers, RGE, and solid dispersion formulations are shown in Figure 5A. Among the hydrophilic carriers, mannitol and P6000 exhibited sharp peaks in a 2θ angle ranging from 10 to 50 and from 15 to 30, respectively. This indicates a typical crystalline structure for both. Consistent with the results, the DSC thermogram showed that mannitol and P6000 had sharp peaks at 166.44 °C and 60.5 °C, respectively (Figure 5B). However, SiO_2_ had a wide peak at around 25°in its XRD pattern, which is attributed to the amorphous structure of SiO_2_ (Aerosil 200). This was also consistent with the DSC thermogram pattern. The XRD and DSC patterns of the RGE freeze-drying powder and the solid dispersion powder of RGE-SiO_2_ also showed amorphous characteristics. A reduction in the diffraction peaks for RGE-M and RGE-P6000 was observed compared with mannitol and P6000, although they still had diffraction peaks, suggesting that the ingredients in the solid dispersion were still in a crystalline state. Similarly, RGE and RGE-SiO_2_ showed no obvious glass transition in the DSC thermogram; however, in the DSC thermogram, RGE-M and RGE-P6000 had endothermic peaks at 144.9 °C and 58.36 °C, respectively (Figure 5B). Taken together, these data suggest that solid dispersion formulations of RGE-M and RGE-P6000 had crystalline structures, whereas RGE and RGE-SiO_2_ stayed in the amorphous state after the freeze-drying process.

#### 3.3.2. Dissolution Rate of Ginsenoside from Solid Dispersion Formulations of RGE

We next investigated the dissolution rates of the solid dispersion formulations of RGE, RGE-SiO_2_, RGE-M, and RGE-P6000 (Figure 6). It was observed that the dissolution patterns for RGE and RGE-M had a similar tendency for all detected ginsenosides GRb1, GRb2, GRc, and GRd. In the solid dispersion formulation containing SiO_2_ and P6000, the initial dissolution rate was lower than that of RGE for 60 min; however, the dissolution rate of all ginsenosides was >85% before the last sampling point (240 min). Considering the crystalline nature of RGE-M and RGE-P6000, it is possible that the lower dissolution rate of RGE-SiO_2_ and RGE-P6000 compared to RGE and RGE-M may be due to the hydrophilicity or wettability of the excipient rather than the crystalline state.

#### 3.3.3. Intestinal Permeability of Ginsenoside from Solid Dispersion Formulations of RGE 

As ginsenoside is a biopharmaceutical classification system (BCS) class III drug [33], it has high solubility but low permeability, which may explain the low bioavailability of the ginsenosides. Therefore, it is pertinent to evaluate the use of carriers and whether they can improve the permeability of solid dispersion formulations of RGE. The intestinal permeability for each formulation is shown in Figure 7. The gastrointestinal permeation of GRb1 from RGE-SiO_2_ was significantly higher than that of RGE; however, GRb1 permeability from RGE-M and RGE-P6000 significantly decreased by 79.1% and 73.1%, respectively, compared with that from RGE. The permeability of GRb2, GRc, and GRd from the solid dispersion formulations of RGE was similar to that of GRb1.

### 3.4. Pharmacokinetics of Ginsenosides in RGE-SiO_2_

Based on the data from our XRD and DSC patterns, the dissolution rate, and the intestinal permeability of ginsenosides from of the solid dispersion formulations of RGE (RGE-SiO_2_, RGE-M, and RGE-P6000), we selected RGE-SiO_2_ as the final formulation. RGE-SiO_2_ had an improved hygroscopic nature of RGE and enhanced the intestinal permeability of ginsenoside. We next investigated the comparative pharmacokinetics of ginsenoside after oral administration of RGE and RGE-SiO_2_ (equivalent to 375 mg/kg as RGE) (Figure 8).

After oral administration of RGE-SiO_2,_ the plasma concentration–time profiles of GRb1, GRb2, GRc, and GRd were higher than those in the RGE group. Consequently, the maximum plasma concentrations (C_max_) and area under the plasma concentration curve (AUC) were significantly greater for GRb1, GRb2, GRc, and GRd in the RGE-SiO_2_ group compared to those in the RGE group without significant alterations in T_max_, T_1/2_, and MRT values (Table 3). The relative bioavailability of GRb1, GRb2, GRc, and GRd in the RGE-SiO_2_ group was 164–183% compared to the RGE group. These data suggest there was an increased absorptive amount of ginsenosides following the administration of RGE-SiO_2_ formulation compared with RGE alone. This is attributed to the increased intestinal permeability of ginsenosides as the formulation of RGE-SiO_2_.

Taken together, the formulation of the solid dispersion of RGE with amorphous SiO_2,_ the powder formulation of RGE, successfully resulted in improved hygroscopicity, increased intestinal permeability, and enhanced oral bioavailability.

### 3.5. Stability of RGE-SiO_2_

Since the solid dispersion formulation usually becomes an amorphous solid powder, it is prone to be recrystallized by increasing the overall molecular mobility, decreasing glass transition temperature, or disrupting interaction between the drug and carrier, resulting in a decreased dissolution rate during the storage period [15,16]. Therefore, we evaluated the stability of the RGE-SiO_2_ formulation by monitoring the XRD patterns, DSC thermograms, ginsenosides content, and dissolution profiles of major ginsenosides from an RGE-SiO_2_ formulation stored for 1 year and compared it with a freshly prepared RGE-SiO_2_ formulation. The XRD and DSC patterns of the RGE-SiO_2_ formulation stored for 1 year showed amorphous characteristics and showed no obvious difference between the two RGE-SiO_2_ formulations regardless of the storage period (Figure 9A,B). The ginsenoside content showed no statistical difference between the two RGE-SiO_2_ formulations regardless of the storage period. The *p* value was >0.05 when compared with freshly prepared RGE-SiO_2_ formulation by the Student *t*-test (Figure 9C). When comparing the dissolution profiles of GRb1, GRb2, GRc, and GRd from the RGE-SiO_2_ formulation stored for 1 year with freshly prepared RGE-SiO_2_ formulation, the dissolution profiles of the four ginsenosides rapidly increased for 60 min, gradually increased for 180 min, and reached more than 85% dissolution before 240 min. The *f*_2_ values of GRb1, GRb2, GRc, and GRd calculated from the mean dissolution rate at time point from 5 min to 180 min resulted in > 50 (Figure 9D). These results suggested that two RGE-SiO_2_ formulations have similar dissolution profiles, and the RGE-SiO_2_ formulation stored for 1 year was stable and contained similar characteristics in terms of crystalline state, ginsenoside content, and dissolution profile from the long-term storage.

## 4. Discussion

In this study, we aimed to identify a carrier that will improve the hygroscopicity of RGE among commonly used hydrophilic carriers. Among various methods for powdering RGE, we utilized a lyophilization method to prepare a solid dispersion formulation, as the solid dispersion formulation is easy to formulate, with minimum use of pharmaceutical excipients, and widely used to enhance oral bioavailability [11,13,15,34]. Among the 10 hydrophilic carriers used in this study, including saccharides (i.e., mannitol, sucrose, and lactose), cellulose derivatives (i.e., HPC), polymers (i.e., PVP, and cross-linked PVP), and polyethylene glycols (i.e., P1500 and P6000), the solid dispersion formulation containing mannitol, PEG6000, and SiO_2_ was most effective at reducing the water sorption rate and showed reversible sorption–desorption isotherms (Figure 2 and Figure 4). However, visual observations of solid dispersion formulations of RGE-SiO_2_, RGE-M, and RGE-P6000 stored at 30 °C with varying relative humidity for 25 days demonstrated different results. Caking was observed in the formulations of RGE-M and RGE-P6000 compared with RGE-SiO_2_. Previously, it has been reported that the use of SiO_2_ (Aerosil 200) increased the flowability and stability of solid dispersion formulations due to its amorphous nature, and it provides a large surface area for adsorption and limits the recrystallization process of RGE [35,36]. In the formulations containing mannitol and P6000 (i.e., RGE-M and RGE-P6000), a return to the powder state was observed when a force was applied with a mortar pestle. RGE-M and RGE-P6000 had partial transformation to an amorphous structure as determined by the XRD and DSC results (Figure 5).

An in vitro dissolution test and in vitro intestinal permeability test were utilized for the characterization of the selected solid dispersion formulations of RGE (RGE-SiO_2_, RGE-M, and RGE-P6000). For this, we measured the concentrations of GRb1, GRb2, GRc, and GRd, which were identified as ginsenosides from plasma samples of mice, rats, and humans following single or repeated oral administrations of RGE [4,5,10,17,18,27,37,38]. In RGE, GRb1, GRb2, GRc, and GRd are all PPD-type ginsenosides present in high quantity (Table 1). Additionally, GRg3, GRe, and GRh1 were found at high content in RGE; however, these ginsenosides were not detectable in rat plasma after oral administration of RGE [5]. In addition, GRb1, GRb2, GRc, and GRd are glycones that have three or four sugar structures. Because of this highly glycosylated structure, GRb1, GRb2, GRc, and GRd are classified as BCS class III (high solubility and low permeability) [33]. Therefore, screening the solid dispersion formulation of RGE as a strategy for enhancing the bioavailability of ginsenoside could be justified. In addition, increasing oral bioavailability is widely achieved using the solid dispersion technique through the enhancement of its dissolution and intestinal permeability of drugs [13], and therefore this technique could be used for enhancing the oral bioavailability of RGE.

The carrier type affected the initial dissolution pattern of solid dispersion formulations of RGE. On average, RGE powder had the fastest initial dissolution rate, and RGE-SiO_2_ showed the lowest initial rate; however, after 4 h, all ginsenosides had >85% dissolution rate, irrespective of the formulations tested (Figure 6). Ginsenosides are highly water-soluble substances with many sugar moieties attached; therefore, a significant improvement of the dissolution by the addition of a hydrophilic carrier does not seem to occur. With the presence of surface silanol groups, SiO_2_ may have been able to form hydrogen bonds with drug molecules during the formulation of solid dispersion, which resulted in a decrease in the initial dissolution [30,36], which was also observed in this study. However, GRb1, GRb2, GRc, and GRd had a slow absorption rates in vivo, and, consequently, plasma ginsenosides resulted in the late T_max_ of about 4–6 h (Figure 8 and Table 3). Therefore, the decreased initial dissolution rate of ginsenosides from RGE-SiO_2_ may not result in a decrease in the in vivo intestinal absorption of GRb1, GRb2, GRc, and GRd.

In addition, RGE-SiO_2_ increased the ginsenoside intestinal permeability of GRb1, GRb2, GRc, and GRd in the rat jejunal segment, whereas RGE-M and RGE-P6000 decreased the ginsenoside permeability (Figure 7). Mannitol and PEG, widely used hydrophilic carriers for solid dispersion formulations, are osmotically active excipients. They increase the osmotic pressure for drugs corresponding to BCS class III, thereby increasing the intestinal fluid and decreasing the intestinal retention time of the substance [39,40]. The use of increasing amounts of mannitol and sorbitol decreased the intestinal absorption of cimetidine and ranitidine, BCS class III drugs, in a dose-dependent manner [39,41,42]. The intestinal absorption of monosaccharides D-xylose and L-rhamnose decreased with the increase in the osmotic load of mannitol [42]. Since GRb1, GRb2, GRc, and GRd are glycosylated ginsenosides and penetrate the intestinal membrane through passive diffusion [18], the presence of mannitol as a carrier could decrease the intestinal permeation of these ginsenosides. A similar phenomenon was also reported in the case of PEG400 and PEG4000 [39]. There is a negative linear relationship between the osmotic potential of PEG4000 and the oral absorption of low-permeability drugs such as chlorthalidone, amoxicillin, and digoxin [43,44,45]. Therefore, the reduced permeation of GRb1, GRb2, GRc, and GRd in RGE-P6000 may share similar properties of osmotic action with PEG4000 and mannitol.

Matsuo et al. [46] reported the increased skin permeation by SiO_2_ (Aerosil 200) without toxicological effect. This result suggested the increased permeability of the drug in the presence of SiO_2_ as a carrier, which may be consistent with our enhanced permeation of GRb1, GRb2, GRc, and GRd from RGE-SiO_2_ compared with that from RGE (Figure 7). In addition to the nature of SiO_2_ as an adsorbent and stabilizer of drugs, the permeation enhancer of poorly permeable BCS class III drugs could provide significant use to SiO_2_. The decreased permeation of ginsenosides from RGE-M and RGE-P6000 formulation, but an increase in permeation of ginsenosides from RGE-SiO_2_, led us to investigate the in vivo pharmacokinetics of ginsenosides using RGE-SiO_2_ formulation. The blood concentration of GRb1, GRb2, GRc, and GRd in rats after oral administration of RGE-SiO_2_ formulation agreed with the results of the previous intestinal permeation experiment.

In accordance with the permeation-enhancing effect of SiO_2_, various pharmaceutical excipients have emerged, not only as solubilizing agents but also as permeation enhancers. For examples, pluronic triblock copolymers such as pluronic F127 and pluronic P85 increased the intestinal permeation of morin and berberine and enhanced their oral bioavailability [47,48]. The use of tween 80, vitamin E-D-α-tocopheryl polyethylene glycol 1000 succinate (TPGS), and polyethylene glycol 400 (PEG400) increased the oral bioavailability of digoxin, curcumin, and silymarin through the increased permeation of these drugs by inhibiting the intestinal first-pass effect of drugs [13,34,49,50,51].

We compared the pharmacokinetic parameters of ginsenosides, and the C_max_ and AUC values of GRb1, GRb2, GRc, and GRd from the RGE-SiO_2_ group were significantly greater than those from RGE groups (using four rats per group). During the post hoc power analysis using our ginsenoside pharmacokinetic data from RGE and RGE-SiO_2_ groups [52], the statistical power values of the existing AUC results were calculated as 84.3%, 80.1%, 89.2%, and 30% for GRb1, GRb2, GRc, and GRd, respectively, with a significance level of 0.05. To determine the minimum number of rats for adequate study power (significance level of 0.05 and statistical power of 80%) of our ginsenoside pharmacokinetic data from RGE and RGE-SiO_2_ groups [53], the minimum sample size was estimated to two to three rats for GRb1, GRb2, and GRc and to six rats for GRd. The estimation indicated that our study design was adequate to detect statistical significance of C_max_ and AUC values of GRb1, GRb2, and GRc that are abundant in RGE and detected in plasma samples. However, we should note that the statistical power for the analysis for GRd was lower than 80% in the pharmacokinetics studies using four rats, which could be attributed higher variability in the plasma concentrations of GRd because of the variable intestinal metabolism of GRd [18]; therefore, it is necessary to expand the number of subjects for the future formulation studies using GRd.

## 5. Conclusions

We successfully formulated the solid dispersion of RGE using SiO_2_ (Aerosol 200) with high flowability, improved hygroscopicity, and enhanced oral bioavailability using a freeze-drying method. The RGE-SiO_2_ formulation had a reversible water sorption–desorption isothermogram and amorphous structure. The intestinal permeability was enhanced and, consequently, improved the oral bioavailability of PPD-type ginsenosides detectable in the plasma samples. In addition, the RGE-SiO_2_ formulation stored for 1 year stayed in an amorphous state and had a similar ginsenoside content and dissolution profile after long-term storage at 25 °C and 30%RH for 1 year compared with the freshly prepared RGE-SiO_2_ formulation. Collectively, the flowable and reduced hygroscopic RGE-SiO_2_ formulation would be helpful to process solid dose formulations of RGE products. This study is the first to demonstrate increased intestinal permeability of solid dispersion formulations of RGE containing SiO_2_.

## Figures and Tables

**Figure 1 pharmaceutics-13-01022-f001:**
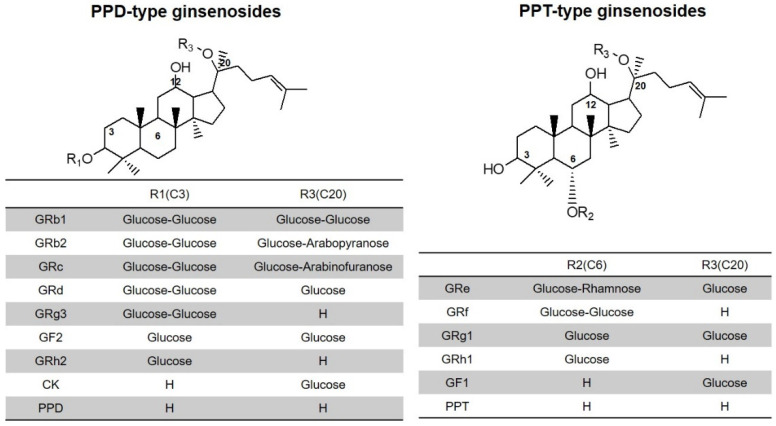
Structure of 20(*S*)-protopanaxadiol- and 20(*S*)-protopanaxatriol-type ginsenosides. Compound K, CK; protopanaxadiol, PPD; protopanaxatriol, PPT.

**Figure 2 pharmaceutics-13-01022-f002:**
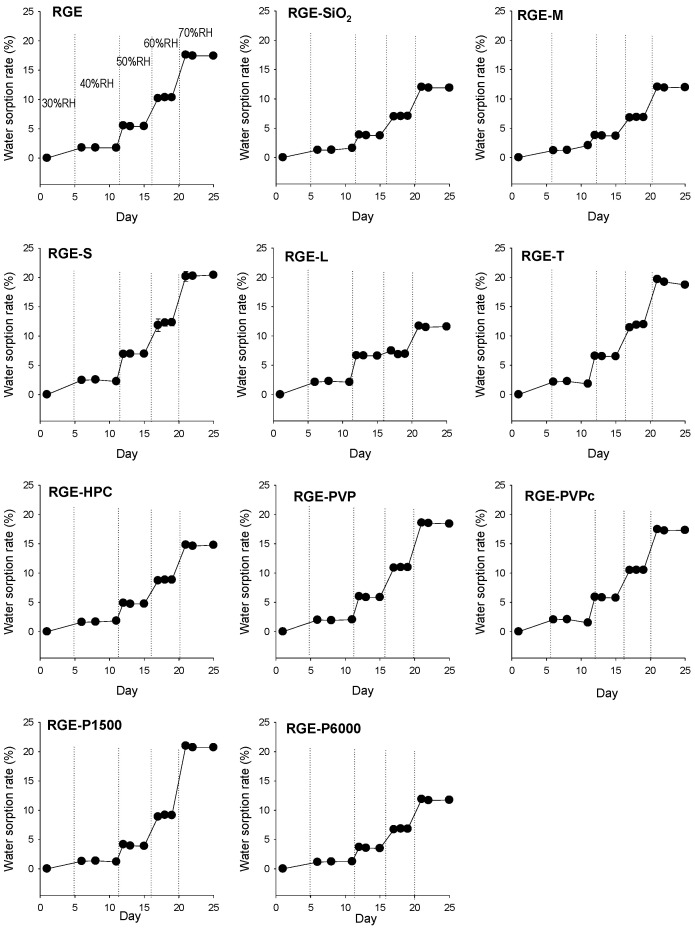
The effect of humidity (from 30% RH to 70% RH) on the water sorption rate of solid dispersion of red ginseng extract (RGE) stored in a thermo-hygrostat at 30 °C. Data represent the mean ± standard deviation (*n* = 3). RGE: solid dispersion of red ginseng extract; RGE-SiO_2_: solid dispersion of RGE with silicon dioxide; RGE-M: solid dispersion of RGE with mannitol; RGE-S: solid dispersion of RGE with sucrose; RGE-L: solid dispersion of RGE with lactose; RGE-T: solid dispersion of RGE with trehalose; RGE-HPC: solid dispersion of RGE with hydroxypropyl cellulose; RGE-PVP: solid dispersion of RGE with polyvinyl pyrrolidone; RGE-PVPc: solid dispersion of RGE with cross-linked polyvinyl pyrrolidone; RGE-P1500: solid dispersion of RGE with PEG1500; RGE-P6000: solid dispersion of RGE with PEG6000.

**Figure 3 pharmaceutics-13-01022-f003:**
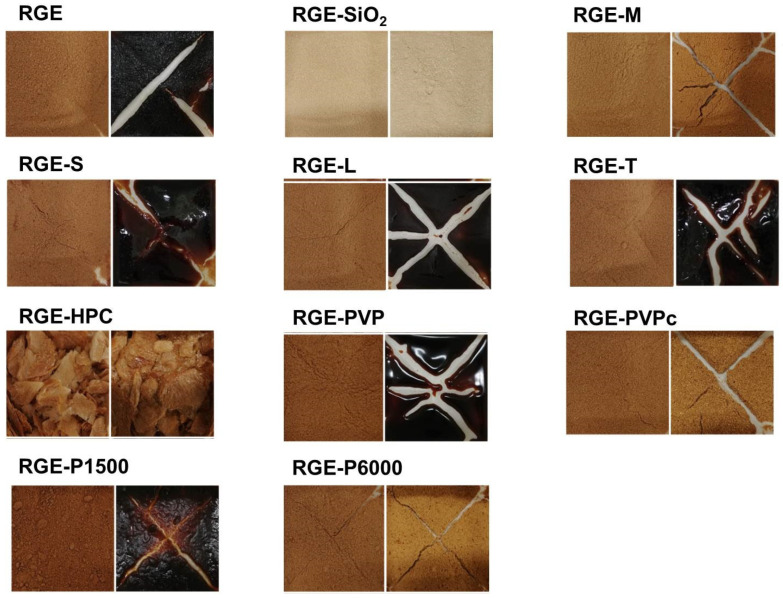
Hygroscopic evaluation of the solid dispersion of red ginseng extract (RGE) with various hydrophilic carriers before (left) and after (right) storing in a thermo-hygrostat incubator with increasing humidity (from 30% RH to 70% RH) for 25 days at 30 °C. RGE: solid dispersion of red ginseng extract; RGE-SiO_2_: solid dispersion of RGE with silicon dioxide; RGE-M: solid dispersion of RGE with mannitol; RGE-S: solid dispersion of RGE with sucrose; RGE-L: solid dispersion of RGE with lactose; RGE-T: solid dispersion of RGE with trehalose; RGE-HPC: solid dispersion of RGE with hydroxypropyl cellulose; RGE-PVP: solid dispersion of RGE with polyvinyl pyrrolidone; RGE-PVPc: solid dispersion of RGE with cross-linked polyvinyl pyrrolidone; RGE-P1500: solid dispersion of RGE with PEG1500; RGE-P6000: solid dispersion of RGE with PEG6000.

**Figure 4 pharmaceutics-13-01022-f004:**
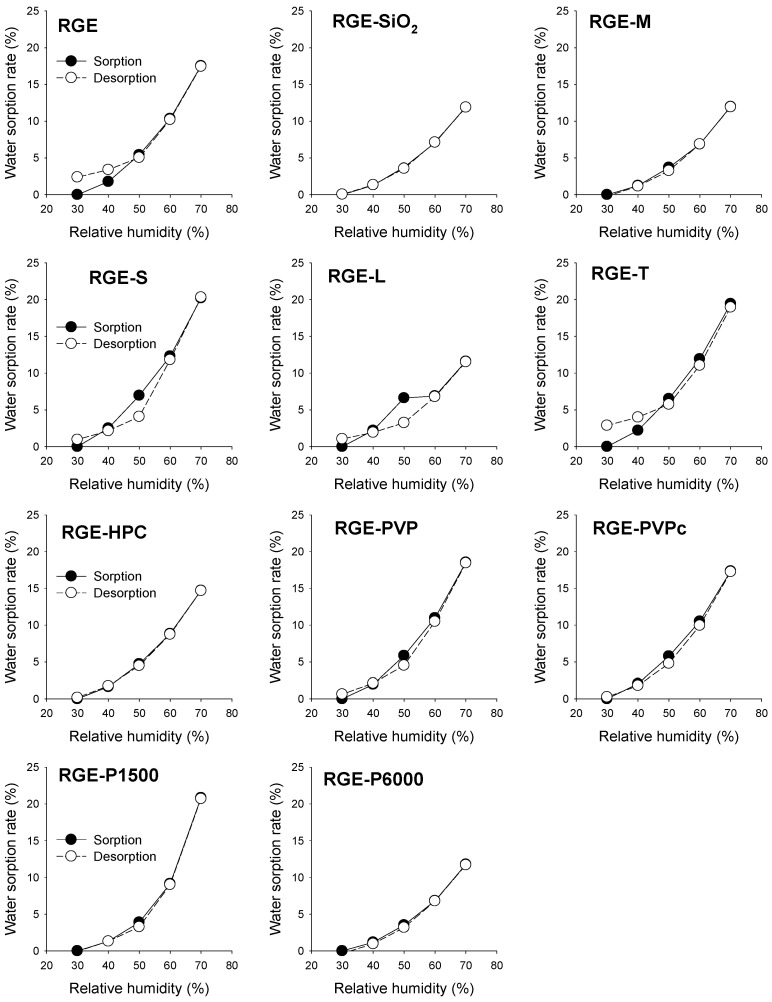
Water sorption–desorption rate curves of solid dispersion formulations of red ginseng extract (RGE) stored in a thermo-hygrostat with increasing relative humidity (from 30% RH to 70% RH; ●) and decreasing relative humidity (from 70% RH to 30% RH; ○) at 30 °C. Data represent the mean ± standard deviation (*n* = 3). RGE: solid dispersion of red ginseng extract; RGE-SiO_2_: solid dispersion of RGE with silicon dioxide; RGE-M: solid dispersion of RGE with mannitol; RGE-S: solid dispersion of RGE with sucrose; RGE-L: solid dispersion of RGE with lactose; RGE-T: solid dispersion of RGE with trehalose; RGE-HPC: solid dispersion of RGE with hydroxypropyl cellulose; RGE-PVP: solid dispersion of RGE with polyvinyl pyrrolidone; RGE-PVPc: solid dispersion of RGE with cross-linked polyvinyl pyrrolidone; RGE-P1500: solid dispersion of RGE with PEG1500; RGE-P6000: solid dispersion of RGE with PEG6000.

**Figure 5 pharmaceutics-13-01022-f005:**
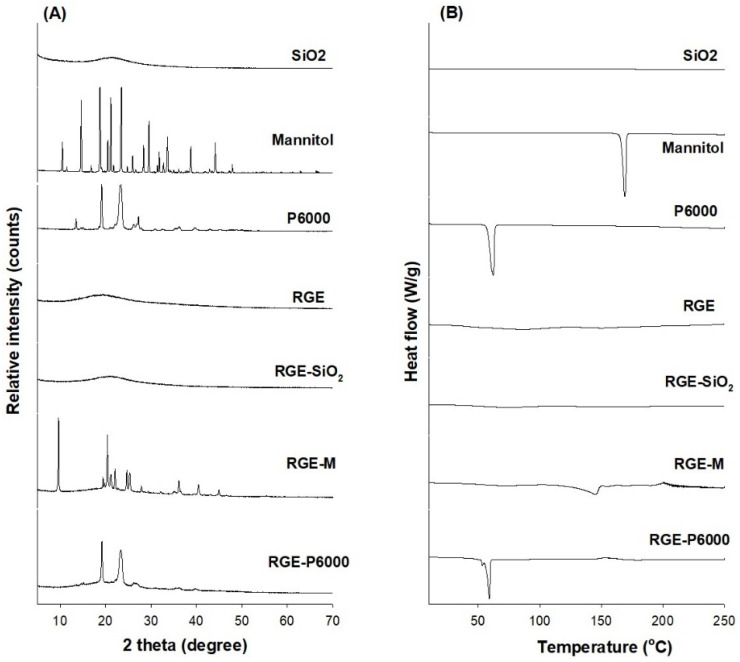
(**A**) The X-ray diffraction patterns and (**B**) the differential scanning calorimetry thermograms of SiO_2_, mannitol, P6000, red ginseng extract (RGE), RGE-SiO2, RGE-M, and RGE-P6000.

**Figure 6 pharmaceutics-13-01022-f006:**
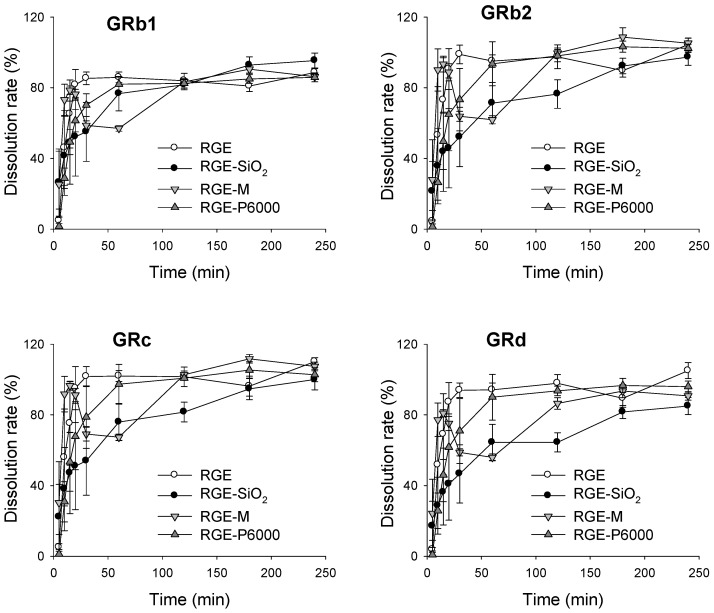
Dissolution profiles of ginsenoside Rb1 (GRb1), GRb2, GRc, and GRd from the solid dispersions of red ginseng extract (RGE), RGE-SiO_2_, RGE-M, and RGE-P6000 were measured using the paddle method for 240 min. Data represent the mean ± standard deviation (*n* = 3).

**Figure 7 pharmaceutics-13-01022-f007:**
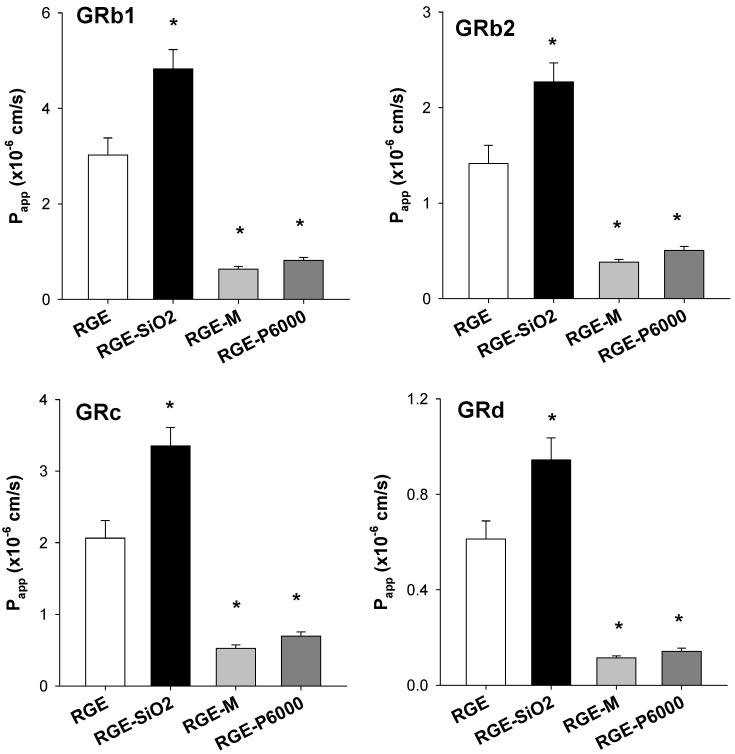
Intestinal permeability (P_app_) of ginsenoside Rb1 (GRb1), GRb2, GRc, and GRd from the solid dispersions of red ginseng extract (RGE), RGE-SiO_2_, RGE-M, and RGE-P6000 was measured in the rat jejunum using the Ussing system. Data represent the mean ± standard deviation (*n* = 3). * *p* < 0.05, statistically significant compared with the RGE group by the Student *t*-test.

**Figure 8 pharmaceutics-13-01022-f008:**
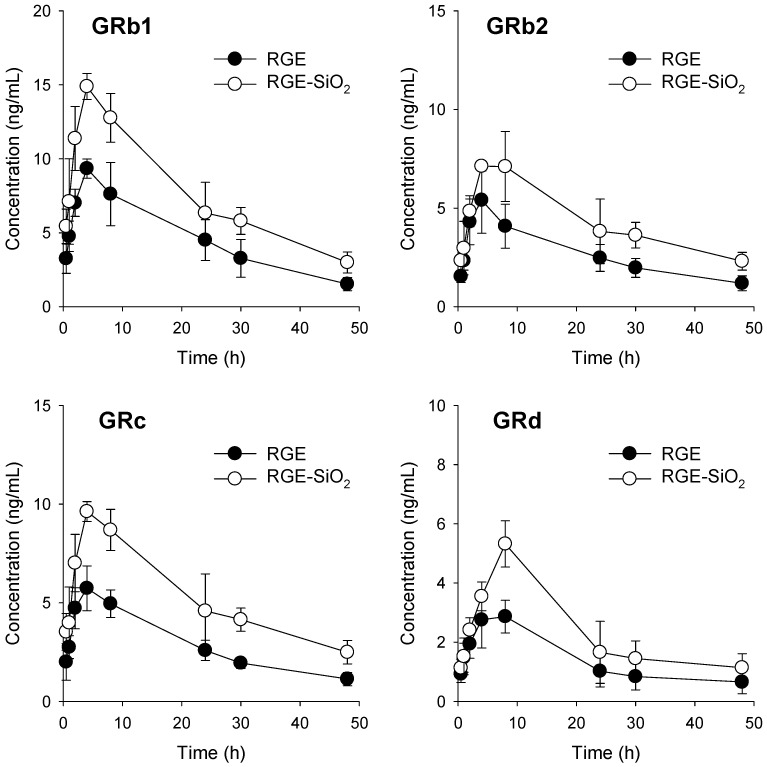
Plasma concentrations of ginsenoside Rb1 (GRb1), GRb2, GRc, and GRd after single oral administration of RGE (375 mg/kg) and solid dispersion formulation of RGE-SiO_2_ (375 mg/kg as RGE) in rats. Data points represent the mean ± SD (*n* = 4).

**Figure 9 pharmaceutics-13-01022-f009:**
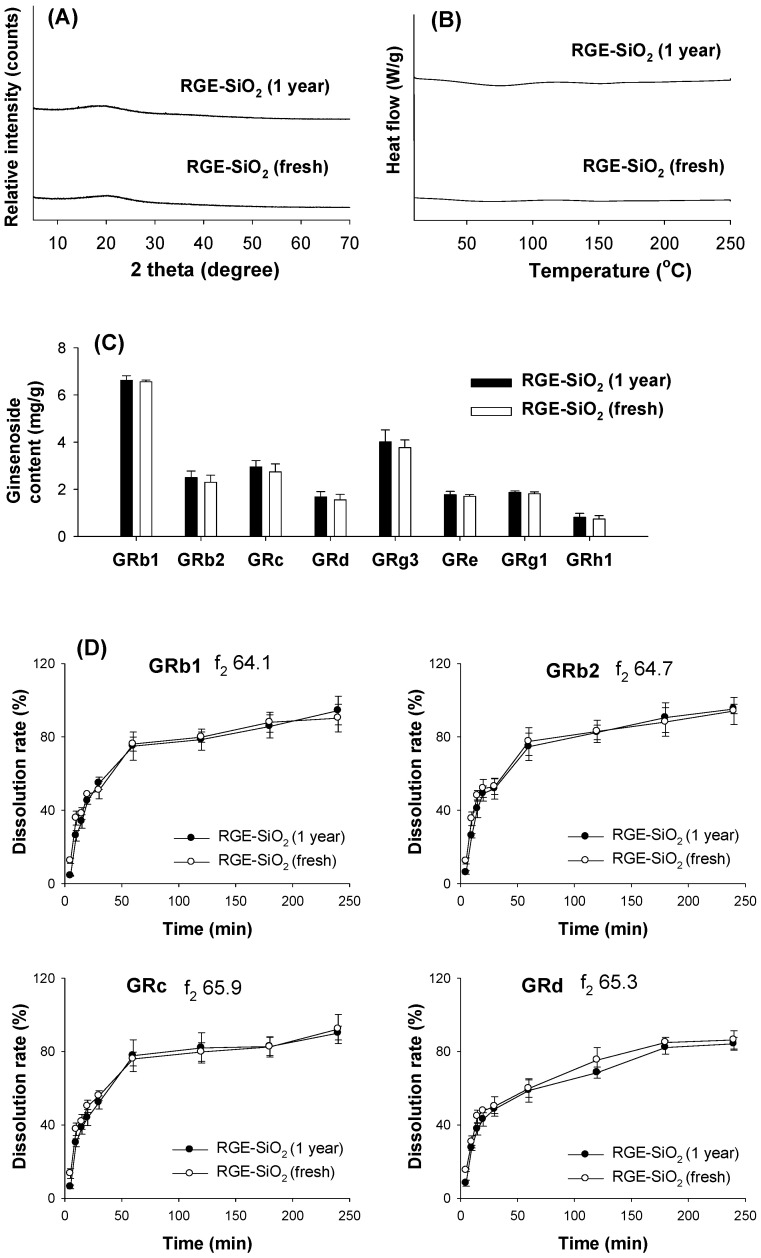
(**A**) The X-ray diffraction patterns and (**B**) the differential scanning calorimetry thermogram of the solid dispersion of RGE-SiO_2_ stored for 1 year compared with freshly prepared RGE-SiO_2_. (**C**) Ginsenoside content detected in the solid dispersion of RGE-SiO_2_ stored for 1 year compared with freshly prepared RGE-SiO_2_. Data represent the mean ± standard deviation (*n* = 4). (**D**) Dissolution profiles of ginsenoside GRb1, GRb2, GRc, and GRd from the solid dispersion of RGE-SiO_2_ stored in thermo-hygrostat (25 °C, 30% RH) for 1 year compared with freshly prepared RGE-SiO_2_ were measured using the paddle method for 240 min. Data represent the mean ± standard deviation (*n* = 12), and the similarity factor (*f*_2_) was calculated from the mean dissolution values of GRb1, GRb2, GRc, and GRd at time points from 5 min to 180 min.

**Table 1 pharmaceutics-13-01022-t001:** Mass spectrometry (MS)/MS parameters for the detection of the ginsenosides and internal standard.

Compound	Retention Time (min)	Adduct	Precursor Ion (*m*/*z*)	Product Ion (*m*/*z*)	Fragmentor Voltage (V)	Collision Energy (V)
GRb1	3.4	[M+Na]^+^	1131.6	365.1	165	65
GRb2	4.2	[M+Na]^+^	1101.6	335.1	185	60
GRc	3.6	[M+Na]^+^	1101.6	335.1	185	60
GRd	4.8	[M+Na]^+^	969.9	789.5	170	50
GRe	1.7	[M+Na]^+^	969.9	789.5	170	50
GRg1	1.8	[M+Na]^+^	824.0	643.6	135	40
GF2	5.9	[M+Na]^+^	807.5	627.5	135	40
GRg3	5.8	[M+Na]^+^	807.5	365.2	165	60
GRh1	3.2	[M-2H_2_O+H]^+^	603.4	423.4	135	10
GF1	3.7	[M+Na]^+^	661.5	203.1	185	40
GRh2	6.6	[M-2H_2_O+H]^+^	587.4	407.4	135	15
CK	6.6	[M+Na]^+^	645.5	203.1	160	35
PPT	5.4	[M-2H_2_O+H]^+^	425.3	109.1	130	30
PPD	7.3	[M-2H_2_O+H]^+^	411.3	109.1	125	25
berberine (IS)	3.5	[M+H]^+^	336.1	320.0	135	30

**Table 2 pharmaceutics-13-01022-t002:** Ginsenoside content determined in the solid dispersion of red ginseng extract.

Ginsenosides		Ginsenoside Content (mg/g)									
		RGE	RGE-SiO_2_ (*p* Value)		REG-M (*p* Value)		RGE-S (*p* Value)		RGE-L (*p* Value)		RGE-T (*p* Value)
PPD-type	GRb1	6.15 ± 2.41	5.58 ± 0.22 (0.651)		5.73 ± 1.06 (0.760)		5.95 ± 1.15 (0.883)		6.36 ± 1.10 (0.883)		6.25 ± 0.89 (0.941)
	GRb2	2.96 ± 1.03	2.53 ± 0.41 (0.462)		2.93 ± 0.48 (0.951)		3.03 ± 0.58 (0.907)		3.11 ± 0.35 (0.907)		3.27 ± 0.64 (0.634)
	GRc	3.35 ± 1.02	2.85 ± 0.40 (0.398)		3.23 ± 0.13 (0.819)		3.52 ± 0.72 (0.794)		3.35 ± 0.17 (0.794)		3.44 ± 0.41 (0.874)
	GRd	1.74 ± 0.79	1.41 ± 0.18 (0.455)		1.63 ± 0.12 (0.806)		1.81 ± 0.43 (0.873)		1.87 ± 0.26 (0.873)		1.73 ± 0.33 (0.981)
	GRg3	3.86 ± 2.19	3.31 ± 0.96 (0.663)		3.48 ± 0.98 (0.759)		3.92 ± 1.61 (0.968)		3.83 ± 1.28 (0.968)		3.84 ± 1.37 (0.987)
PPT-type	GRe	1.71 ± 0.63	1.74 ± 0.53 (0.941)		1.72 ± 0.19 (0.984)		1.79 ± 0.37 (0.836)		1.74 ± 0.21 (0.836)		1.79 ± 0.17 (0.810)
	GRg1	2.11 ± 0.69	1.86 ± 0.41 (0.546)		2.00 ± 0.10 (0.757)		2.12 ± 0.46 (0.977)		2.20 ± 0.06 (0.977)		2.16 ± 0.21 (0.892)
	GRh1	0.71 ± 0.22	0.61 ± 0.07 (0.432)		0.70 ± 0.08 (0.996)		0.73 ± 0.11 (0.869)		0.73 ± 0.07 (0.869)		0.76 ± 0.09 (0.678)
**Ginsenosides**		**Ginsenoside Content (mg/g)**									
		**RGE-HPC (*p* Value)**		**RGE-PVP (*p* Value)**		**RGE-PVPc (*p* Value)**		**RGE-P1500 (*p* Value)**		**RGE-P6000 (*p* Value)**	
PPD-type	GRb1	6.48 ± 1.31 (0.821)		6.08 ± 1.06 (0.960)		6.33 ± 0.89 (0.894)		6.44 ± 0.64 (0.828)		6.49 ± 1.29 (0.836)	
	GRb2	3.12 ± 0.48 (0.793)		2.96 ± 0.58 (0.998)		3.24 ± 0.49 (0.640)		3.25 ± 0.40 (0.622)		2.98 ± 0.28 (0.978)	
	GRc	3.31 ± 0.46 (0.943)		3.46 ± 0.58 (0.859)		3.40 ± 0.12 (0.918)		3.53 ± 0.30 (0.740)		3.62 ± 0.24 (0.678)	
	GRd	1.87 ± 0.34 (0.767)		1.69 ± 0.21 (0.906)		1.75 ± 0.26 (0.978)		1.72 ± 0.18 (0.973)		1.82 ± 0.24 (0.878)	
	GRg3	3.76 ± 1.46 (0.941)		3.67 ± 1.42 (0.888)		3.89 ± 1.28 (0.981)		3.59 ± 1.27 (0.840)		3.89 ± 1.38 (0.980)	
PPT-type	GRe	1.68 ± 0.28 (0.925)		1.74 ± 0.28 (0.924)		1.74 ± 0.20 (0.939)		1.81 ± 0.32 (0.796)		1.71 ± 0.25 (0.995)	
	GRg1	2.25 ± 0.51 (0.758)		2.18 ± 0.39 (0.869)		2.13 ± 0.27 (0.954)		2.22 ± 0.38 (0.793)		2.16 ± 0.39 (0.910)	
	GRh1	0.77 ± 0.13 (0.653)		0.78 ± 0.07 (0.565)		0.73 ± 0.08 (0.810)		0.77 ± 0.05 (0.611)		0.74 ± 0.10 (0.831)	

Among 14 ginsenosides tested, GRh2, GF2, CK, PPD, GF1, and PPT were not detected. Data represent the mean ± standard deviation (*n* = 4). *p* value was provided by comparing ginsenoside content with that in the RGE group using the Student *t*-test.

**Table 3 pharmaceutics-13-01022-t003:** Pharmacokinetic parameters of ginsenoside GRb1, GRb2, GRc, and GRd after single oral administration of RGE (375 mg/kg) and solid dispersion formulation of RGE-SiO_2_ (375 mg/kg as RGE) in rats.

Formulation	Parameters	GRb1	GRb2	GRc	GRd
RGE	C_max_ (ng/mL)	9.5 ± 0.9	5.4 ± 1.7	5.9 ± 1.2	3.2 ± 1.4
	T_max_ (h)	5.0 ± 2.0	4.0 ± 0.0	6.0 ± 2.3	6.0 ± 2.3
	AUC_48h_ (ng × h/mL)	222.5 ± 53.7	127.7 ± 31.6	138.6 ± 30.3	68.6 ± 33.4
	t_1/2_ (h)	16.1 ± 1.5	21.7 ± 3.6	19.2 ± 0.8	25.6 ± 16.4
	MRT (h)	16.8 ± 1.1	17.9 ± 0.9	16.9 ± 0.5	16.2 ± 1.0
RGE-SiO_2_	C_max_ (ng/mL)	15.5 ± 1.9 *	8.1 ± 2.1 *	10.5 ± 1.9 *	5.5 ± 2.6 *
	T_max_ (h)	5.0 ± 2.0	5.0 ± 2.0	5.0 ± 2.0	7.0 ± 2.0
	AUC_48h_ (ng × h/mL)	363.8 ± 78.7 *	209.5 ± 57.7 *	253.5 ± 65.2 *	115.0 ± 50.7 *
	t_1/2_ (h)	20.1 ± 4.2	24.9 ± 4.1	21.3 ± 2.8	28.9 ± 22.0
	MRT (h)	17.3 ± 0.7	19.1 ± 0.6	18.2 ± 0.6	16.6 ± 0.6
	Relative BA (%)	163.5	164.1	182.9	167.7

C_max_, maximum plasma concentration; T_max_: time to reach C_max_; AUC, area under the curve; t_1/2_, half-life; MRT, mean residence time; relative BA, relative bioavailability (AUC_RGE-SiO__2_/AUC_RGE_ × 100). * *p* < 0.05, statistically significant compared with RGE group by the Student *t*-test. Data represent the means ± standard deviation (*n* = 4).

## Data Availability

The data presented in this study are available upon request.

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
