# Peer review of "Improved Hygroscopicity and Bioavailability of Solid Dispersion of Red Ginseng Extract with Silicon Dioxide"

_pharmaceutics, 2021, doi:10.3390/pharmaceutics13071022_

Round 1

Reviewer 1 Report

In this work, the author developed a powder of the Korean red ginseng 17 extract (RGE)-based solid dispersion. The concept of study seems to follow the conventional herbal-based supplements. However, the content of the manuscript has poor scientific discussions in terms of pharmaceutics. Additionally, the quality of the submitted work seems to be no in-depth understanding for satisfying the criteria of pharmaceutical drug development. Therefore, I would reject and encourage the resubmission of the manuscript.

  1. The authors should present a detailed description for the preparation and characterization of red ginseng extract. The extraction efficiency of red ginseng extract accordingly should be accompanied by a lab-scale procedure for reproducibility. Additionally, the recovery and identification of active pharmaceutical ingredients in the extract should be validated by considering quality control. A thorough comparison will doubtless improve the discussion extent of the current research article related to herbal extract-based drug development (https://doi.org/10.1080/10717544.2020.1716876, https://doi.org/10.1080/10837450.2019.1703134, https://doi.org/10.1016/j.ejps.2019.105204)
  2. In order to further support the potential of this formulation as pharmaceutical drug development, it is necessary to explain the standardized experimental method that should follow the US pharmacopeia with the powder properties of the pharmaceutical product, dissolution test, capsule size, and disintegration test.
  3. The stability conditions related to the active pharmaceutical ingredient need to be monitored for 3 batches manufactured under long-term storage conditions and accelerated test conditions, although 100% of the ICH guidelines cannot be followed by these kinds of lab-scale experiments. In this study, the stability results confirmed that the significant stability of the active pharmaceutical ingredient in the form of stress conditions, but it is not sufficient as a stability result that can be supported in terms of quality management.
  4. Fundamental research approaches are needed for the solubility or permeability of active pharmaceutical active ingredients in red ginseng extract. It would be more desirable to obtain meaningful in vivo results based on cellular and in vitro experiments. It is not considered unnecessary and rational experimental design direction to obtain statistically significant results by adding 50% of the inactive ingredient to the extract.
  5. The current direction of pharmaceutical science research is to examine the principle more in a system capable of reproducing a living body based on a certain mechanism rather than a simple in vivo administration of a manufactured formulation, and finally to verify the result through a living body. The present authors have achieved very meaningful results, but the extent of discussion and in-depth mechanism of improvement of bioavailability is poor.

Reviewer 2 Report

This study formulated the solid dispersion of RGE using SiO2 with improved hygroscopicity, and enhanced oral bioavailability. Results of various physical and pharmaceutical tests are well compared and summarized. Major revisions are required before the acceptance. Comments and suggestions are attached below:

  1. Section “Introduction” the authors should introduce more about “solid dispersion technique” including advantages and current background. In addition, the authors should describe that if there is any research in this field related to ginseng.
  2. Line 220 “frequentlyused” should be revised as “frequently used”.
  3. Table 2 The authors should calculate the p values of t test to support results of “no statistical difference”.
  4. It is hard to identify the dissolution rates of different sample before 60 min in figure 6, and thus it is suggested that the icons of different samples should be smaller to identify the trend of dissolution rates.

Reviewer 3 Report

    This work formulates a solid dispersion of red ginseng extract (RGE) with  high adsorptive carriers to improve both the hygroscopicity and oral bioavailability of RGE. The writing of the manuscript is clear. There are some suggestions in terms of experimental design and result discussion.

1. The authors gave the results that “the gastrointestinal 350 permeation of GRb1 from RGE-SiO2 was significantly higher than that of RGE, however, 351 GRb1 permeability from RGE-M and RGE-P6000 significantly decreased by 79.1% and 352 73.1%, respectively, compared with that from RGE”. Please explain why RGE-SiO2 and other RGE solid dispersions give opposite results.

2. The sample size of the pharmacokinetic experiment in rats is four. Please prove that this sample size is large enough.

3. For amorphous solid dispersion, physical stability is one of the most important issue. However, the authors didn’t give the related result. It is necessary to add it.

4. In Table 2, n value should be given.

Reviewer 4 Report

In this paper authors report research regarding solid dispersion formulation of red ginseng extract (RGE) using SiO2 (Aerosol 200) with high flowability, improved hygroscopicity, enhanced intestinal permeability and consequently, improved oral bioavailability. The manuscript is carefully conceived and clearly divided into sections. It is well discussed and presents a nice piece of methodological work. The authors carried out a number of measurements in order to select the most suitable carrier for optimization the biopharmaceutical and pharmacokinetic properties of RGE. The conclusions are strongly supported by the experimental work. Therefore, I would like to recommend the publication of the manuscript after minor editorial revision. At the same time I would like to ask the authors to check the reference 18 and 19 presented in line 225 in relation with excipients used in the preparation of solid dispersion formulations.

Author Response

We deeply appreciate the reviewer’s positive comments. We carefully checked the typographical errors and references that were mistakenly arranged are corrected during the revision.

Round 2

Reviewer 3 Report

The quality of this manuscript has improved after revision. I recommend it to be published on pharmaceutics after minor revision. I hope the author carefully check all the information in this manuscript to ensure all are correct. For example, in line 97, Nimotop®  is marketed in soft capsules and has nothing to do with solid dispersions (https://www.accessdata.fda.gov/drugsatfda_docs/label/2006/018869s014lbl.pdf). 

Author Response

The quality of this manuscript has improved after revision. I recommend it to be published on pharmaceutics after minor revision. I hope the author carefully check all the information in this manuscript to ensure all are correct. For example, in line 97, Nimotop®  is marketed in soft capsules and has nothing to do with solid dispersions (https://www.accessdata.fda.gov/drugsatfda_docs/label/2006/018869s014lbl.pdf). 

Answer> We thank you for the valuable comments that improve the quality of our manuscript. As the reviewer suggested, we carefully checked our manuscript and corrected our mistakes. Especially, we deleted Nimotop and Torcetrapib because Nimotop is soft gelatin capsule formulated using glycerin, peppermint oil, purified water, and polyethylene glycol 400 mixture as a vehicle and the development of torcetrapib was terminated during the clinical phase III. Instead, we added another representative marketed products based on solid dispersion techniques.

[Page 3, line 95] a number of commercial solid dispersion products including Cesamet® (nabilone-PVP; Lilly), Ceritan® (everolimus-HPMC; Norvatis), Gris-PEG® (griseofulvin-PEG; Norvatis), Intelence® (etravirine-HPMC; Janssen), Adalat-XL® (nifedipine-PEG3350/HPC/cellulose acetate; Bayer), Sporanox® (itraconazole-HPMC; Janssen), Isoptin SR® (verapamil-HPC/HPMC; Abbott), and Crestor® (rosuvastatin-HPMC; Astrazeneca) etc. were marketed as tablet or capsule formulation [15,16].

[References]

  1. Tran, P.; Pyo, Y.C.; Kim, D.H.; Lee, S.E.; Kim, J.K.; Park, J.S. Overview of the manufacturing methods of solid dispersion technology for improving the solubility of poorly water-soluble drugs and application to anticancer drugs. Pharmaceutics 2019, 11, E132. doi:10.3390/pharmaceutics11030132.

16. Cid, A.G.; Simonazzi, A.; Palma, S.D.; Bermúdez, J.M. Solid dispersion technology as a strategy to improve the bioavailability of poorly soluble drugs. Ther. Deliv. 2019, 10, 363-382, doi:10.4155/tde-2019-0007.